# Consumers’ Attitude towards Supermarket and Proximity Stores as Purchasing Outlets of Italian Potato Consumers

**DOI:** 10.3390/foods12152877

**Published:** 2023-07-28

**Authors:** Antonella Samoggia, Giulia Rossi, Federica Beccati

**Affiliations:** Department of Agricultural and Food Sciences, University of Bologna, Viale Fanin, 50, 40127 Bologna, Italy; giulia.rossi109@unibo.it (G.R.);

**Keywords:** consumer, potato, sales channel, behavior, food

## Abstract

World potato consumption has fallen. Similarly, Italian consumers are buying fewer potatoes, despite the high number of certified quality and innovative potatoes being produced. Moreover, Italian consumers’ knowledge of potato characteristics and innovations tends to be limited. To increase consumer engagement and revitalize the market, strategic marketing efforts need to be implemented by addressing the different characteristics of consumers in the different purchase channels. The aim of this study is to explore and differentiate consumer purchasing behavior and attitudes towards potatoes in supermarkets and neighborhood channels. The study collected 855 responses through an online survey. Data processing included the creation of indices of consumers’ knowledge of potato nutrition and their propensity to innovate, as well as logistic regression to analyze the determinants of potato-purchasing behavior. The results show that consumers with increased potato consumption, a higher level of education, and employment prefer to buy potatoes in the supermarket. A preference for potato innovation also increases the likelihood that consumers will buy potatoes in supermarkets. Conversely, consumers with elderly relatives at home and a low level of education, but a high level of knowledge about the nutritional properties of potatoes, prefer to shop through neighborhood channels.

## 1. Introduction

Potatoes have their origin in the Andes and are grown worldwide. As a staple food consumed by billions of people each day, potatoes rank as the world’s third-most important crop. Specifically, potatoes are the world’s number one non-grain food commodity, playing a crucial role in achieving global food security [1,2]. However, in recent decades, there has been a decline in global potato consumption. Several reasons have been identified for this decline, including the growing preference for ready-made foods, the introduction of new food options, and consumers’ increasing inclination towards healthy and nutritious diets [3,4,5]. Moreover, consumers often perceive potatoes as a commodity rather than recognizing their nutritional properties or distinguishing the unique characteristics of different potato varieties. An Australian study revealed a low level of awareness regarding the dietary potential and benefits of potatoes [5]. In the past two decades, the popularity of potatoes has faced further challenges due to declining harvest yields and associated price fluctuations [6]. Nevertheless, contrary to consumer perception, potato production exhibits a high level of variety diversification and continual product innovation. For instance, in Italy, standard potatoes represent only 20% of the total potato offering [7]. Additionally, recent years have witnessed significant growth in the potato innovation sector, contributing to the development of sustainable agri-food systems [8,9]. One such innovation is biofortification, which aims to improve the micronutrient composition of potatoes and has been identified as an effective strategy for reducing micronutrient deficiencies in populations [10].

In order to revitalize the market, it is essential to align the perspectives of potato producers and consumers. Among the attributes that can influence consumer assessment of potatoes, information about different potato characteristics plays a significant role [11]. However, limited marketing efforts have been made to differentiate the numerous varieties of potatoes [12], and little information is provided on the potato package or label [13]. Previous studies have explored factors and values that affect consumer behavior at the point of potato purchase [5,13,14]. However, companies need to understand how consumers’ profiles and attitudes differ when shopping for potatoes in different sales channels in order to make strategic marketing decisions. A research gap exists as most recent literature focuses on factors influencing consumers’ choice between online and physical purchasing channels for fruits and vegetables [15,16] while potatoes are still commonly sold at physical points of sale. This paper examines the drivers of potato purchasing between supermarkets and proximity channels to identify areas for market development.

Potato consumption remains prevalent and valued in the Italian diet, with 69% of Italians regularly consuming fresh potatoes in 2022 [17]. However, there has been an overall decline in per capita potato consumption over the past 10 years [18], and Italians generally exhibit low awareness regarding the nutritional characteristics of potatoes [19]. Italian potato quality is highly regarded, and numerous Italian companies are at the forefront of innovative production techniques. Nonetheless, Italian potato production is relatively modest compared to other EU countries, accounting for only 2.6% of the EU’s harvested production [20]. This limited scale of potato farming poses challenges in terms of price competitiveness, both on a global scale and within the domestic potato market. Consequently, local varieties are primarily sold in local markets rather than nationwide. To revitalize the Italian potato market, an effective marketing strategy may focus on product differentiation by emphasizing the unique properties of local potato varieties [21]. In order to determine which potato features are most advantageous to highlight in each purchasing channel, this study aims to provide various potato-selling outlets with a cross-channel customer segmentation that takes into account consumers’ perceptions, characteristics, and attitudes.

The revitalization of potato popularity in Italy necessitates a comprehensive approach. Informing consumers about the health benefits of potato consumption, as well as the distinctive characteristics and innovations associated with each potato variety, is key. Effective marketing strategies should be built upon consumers’ existing opinions and knowledge regarding potatoes. Furthermore, these strategies should be specifically tailored to the unique characteristics of consumers in different potato-selling environments. A research gap exists in understanding how consumer beliefs and attitudes differ between supermarkets and proximity channels, which are the primary potato-purchasing channels. It is crucial for potato companies to gain insight into consumers’ perspectives and attitudes within each channel while also identifying distinct consumer segments. Therefore, this study aims to explore the characteristics and beliefs that differentiate consumers in various potato-purchasing channels while investigating the role of specific potato knowledge and attitudes towards innovative potato products. The findings of this study will serve as a basis for future advancements in the potato industry and contribute to the formulation of strategic management approaches aimed at enhancing the positioning of potatoes as a healthy and nutritious food product.

## 2. Materials and Methods

Data collection was conducted through an online survey comprising six main sections: purchasing habits, nutritional knowledge, beliefs and attitudes towards potatoes, innovation propensity, and socio-demographic characteristics.

The section on purchasing habits consisted of three questions. One question inquired about consumers’ potato-purchasing channels, while the other two questions focused on the frequency of buying potatoes and the consumption trend over the past five years. To assess nutritional knowledge, two questions were included. Respondents were asked to self-assess their knowledge of the nutritional profile of potatoes and to identify the levels of micronutrients present in potatoes. The formulation of this section received input from a team of nutritionists. The section addressing beliefs and attitudes towards potatoes comprised one question that included several items sourced from common beliefs and attitudes about potatoes, as proposed by Wood et al. (2017) [5]. Innovation inclination was explored through three specific questions concerning respondents’ willingness to purchase enriched potatoes, biofortified potatoes, and potatoes with increased nutrients. The final survey encompassed nine main questions, along with five socio-demographic questions.

The online data collection software adopted is Qualtrics survey programme. The survey employed multiple-choice questions to assess purchasing habits, socio-demographic characteristics, and self-assessment of nutritional knowledge. For measuring innovation inclination, yes-or-no questions were used to determine respondents’ willingness to purchase different innovative products. The remaining questions employed a five-point Likert scale to gauge the level of agreement or disagreement.

The survey was distributed online through major social media platforms using an extensive convenience sampling approach. It specifically targeted Italian respondents due to the study’s focus. The online distribution allowed for wide dissemination across the country over a two-month period, starting in January 2022. The survey was thoughtfully designed to lead respondents through a logical progression, with questions gradually increasing in difficulty and becoming more focused on personal perspectives and ideas. Careful consideration was given to the number and length of questions to ensure that respondents could complete the survey within a reasonable time frame of approximately 10 min.

### Data Elaboration

The data elaboration process comprises three main steps: descriptive analysis, index creation, and model fitting. The first step involves conducting a descriptive analysis to examine the socio-demographic characteristics of the consumer sample as well as to analyze the attitudes, purchasing habits, and consumption patterns of potato consumers.

In the second step, the research developed two separate indexes: One to measure nutritional knowledge and one to express the inclination toward innovative potatoes. Regarding nutritional knowledge, the survey included two questions. First, respondents self-assessed their perceived level of knowledge about potatoes’ nutritional content and second, they were presented with an eleven-item questionnaire to measure their actual knowledge on the subject. For the nutritional assessment, boiled potato was chosen as the reference for measuring nutritional content, considering that potatoes are typically not consumed raw. There are various cooking methods available, yet the choice of the boiled potato as the baseline was supported by recent consumer sensory analysis and nutritionist approaches [22,23]. Boiling potatoes has a longstanding tradition in Italy and is widely recognized as one of the most common cooking methods, making it a familiar concept among the population [24]. Therefore, using boiled potato as the reference ensures relevance and familiarity in assessing the nutritional content of potatoes in this study. The nutritional assessment question asked respondents to indicate the appropriate level of various listed potato micronutrients. For each micronutrient, respondents were required to select a value on a 5-point Likert scale ranging from “Not contained” to “High content,” with the option “I don’t know” available. The study then calculated the difference between the content value selected by each respondent and the actual content value defined by the nutritionist team for each micronutrient, treating the varying content level options as cardinal variables. Table 1 presents the information provided by the nutritionist team. The response option “I don’t know” was considered the maximum possible difference within the Likert scale and, therefore, regarded as equivalent to the maximum possible error, as it reflects the respondents’ lack of knowledge on the subject. To create a “Knowledge Index,” the sum of these differences was normalized and subtracted from 10, resulting in an index ranging from 0 to 10. Finally, the Knowledge Index was correlated with the self-assessment question. After confirming the actual correspondence between perceived and actual knowledge, the study employed perceived knowledge as an independent variable in the model, arguing that consumers make choices based on their perception of reality.

The second index, which pertained to the propensity for innovative potatoes, summarized the responses to three distinct questions. Specifically, the research examined consumers’ willingness to purchase innovative potato products, assessing their inclination towards (i) biofortified potatoes, (ii) enriched potatoes, and (iii) potatoes with increased nutrients, through a series of yes-or-no questions. Due to the semantic similarity and statistical association among the questions, they were consolidated into a single index that ranged from 0 to 3, depending on the number of positive responses provided.

Thirdly, the study employed a logistic regression model to estimate the multivariate association among the collected variables. The dependent variable considered was the potato-purchasing channel. This variable, located in the habits section of the questionnaire, aimed to determine the respondents’ chosen sales channel when purchasing potatoes. Consumers could select one or more of the following channels: supermarkets, local shops, and markets. To create a dichotomous variable, the purchasing channel was transformed into two exclusive levels: consumers who exclusively purchase potatoes from supermarkets and those who exclusively use proximity channels. During the data elaboration phase, the relationships between the variables of interest and the associations between two or more variables were explored. Bivariate analysis was conducted through cross-tabulations and a chi-squared test (χ^2^), revealing various interrelationships among the independent variables. In order to detect possible effect modifiers, the study examined the interactions between pairs of independent variables and the dependent variable. Only one significant relationship was found and included in the model. Additionally, model construction followed a forward selection method, and a multicollinearity test was performed by calculating tolerance and variance inflation factor (VIF) values. The logistic regression model was well-suited for the categorical and dichotomous nature of the dependent variable, i.e., the consumers’ purchasing channel. Unlike multiple regression analysis, logistic regression does not assume a normal distribution for the dependent variable, making it the appropriate method for this type of variable. Furthermore, one of the main focuses of this regression is to classify consumers into different groups: Given a set of independent variables, logistic regression allows the researcher to accurately estimate the likelihood of an individual falling into one of the categories [26]. Among the model parameters, the β values and the significance of each variable were examined. The strength of the association for each variable was measured as an odds ratio (OR) with 95% confidence intervals (CI), indicating the probability of a change in disposition towards the purchasing channels when the variable under consideration changes. Data analysis was performed using IBM SPSS Statistics (version 26, Armonk, NY, USA).

## 3. Results

### 3.1. Sample

After data cleaning and excluding questionnaires with a completion rate below 65%, the final sample consisted of 855 responses. It is worth noting that all respondents in the sample reported consuming potatoes, so no exclusions were made based on this criterion. The sample primarily comprised female participants (78.20%). In terms of age distribution, the majority fell within the 35–54 age range (46.5%), and a significant portion of the respondents were employed (74.5%). Approximately 36.7% of the sample had a university degree, while 41.6% had completed secondary school. Furthermore, 18.4% of respondents reported having an elderly person in their household (Table 2).

### 3.2. Knowledge Level and Innovation Propensity

The results indicate a significant association between consumers’ perceived knowledge and their actual knowledge regarding the nutritional properties of potatoes. The distribution of the knowledge index is centered around the mean, as evident from the percentile results. The mean score is 6.4, with a standard deviation of 1.5, and the range spans from 0 to 10 (Table 3). The minimum score recorded is 0.9, while the maximum score is 9.7. These findings suggest that the overall level of knowledge among the respondents is moderate. The association between perceived and actual knowledge was found to be statistically significant, as evidenced by a χ^2^ test with a significance level of *p* < 0.05. Analyzing the relationship between these two variables reveals that individuals who self-declare as having low knowledge achieve an average score of 5.9, while those with moderate knowledge obtain a score of 6.7. Respondents who perceive themselves as well-informed achieve a higher score of 7.3. This result is encouraging, but it also highlights room for improvement. Consumers tend to behave based on their perceived knowledge, but the correlation implies that implementing an information campaign could be valuable in enhancing both types of knowledge.

When examining the positive responses to individual questions, the results indicated that 64.3% of respondents expressed a willingness to purchase enriched potatoes (Table 4). Biofortified potatoes received positive responses from 56.8% of the participants, while potatoes with increased nutrients scored positively in 46.2% of the cases. Notably, there was a preference for biofortified potatoes over those with increased nutrients, despite the two categories being very similar and overlapping. These findings further underscore the pressing need for a robust information campaign to promote awareness of the potential and innovation surrounding potatoes.

### 3.3. Drivers of Potatoes Purchasing Channels

The results emphasize the high dichotomization of the purchasing channel variable into the two main options. As shown in Table 5, the choices of exclusively purchasing from supermarkets or proximity channels alone accounted for 88.3% of the responses. Only 11.7% of consumers reported using both channels to purchase potatoes. Consequently, the final model was constructed using only the exclusive channel choices, resulting in a sample size of 572 observations.

The binary dependent variable, Yi, represents the purchasing channel and takes the values “Proximity” and “Supermarket.” The probability, P(Y = Supermarket|x), denotes the likelihood of an individual purchasing potatoes in a supermarket, conditioned by the variables included in the questionnaire. The estimated model can be expressed as follows:(1)$$\begin{array}{l} Ln\;\left(Purchasing\;Channel\right)\\ \;\;\;\;\;=\alpha \;+\;\beta 1i\;\;\mathrm{Socio}-\mathrm{Demographics}i+\;\beta 2\;\mathrm{Dietary}\;\mathrm{habits}\;+\;\beta 3\;\mathrm{Nutritional}\;\mathrm{Knowledge}\\ \;\;\;\;\;+\;\beta 4\;\mathrm{Knowledge}\;\mathrm{level}\;\times \;\mathrm{Educational}\;\mathrm{Level}+\;\beta 5\;\mathrm{Opinions}\;+\;\beta 6\;\mathrm{Innovation}\;\mathrm{Predisposition} \end{array}$$

Table 6 presents the results of the fitted model. The analysis for multicollinearity was conducted, and no evidence of multicollinearity was found, as indicated by the highest estimated VIF value of 1.202 and the tolerance levels ranging between 0.832 and 0.971. The final fitted model is statistically significant, with a χ^2^ value of 84.044 and *p* < 0.01. Regarding the significance of variables in the model, the level of knowledge exhibits the highest level of significance, particularly when considering its relationship with the level of education. This finding confirms the importance of knowledge in the study. The variable “Consumption trend” is significant at *p* < 0.05 for the level “Unchanged consumption,” but its significance reduces to *p* < 0.1 for the level “Increased consumption” and becomes insignificant in the last level. Among the tested opinions, two were found to have a significant impact. The belief that compares potatoes to bread instead of vegetables shows significance at *p* < 0.05. Additionally, the opinion that potatoes require a long cooking time demonstrates significance at *p* < 0.1. The *p*-value representing the variable “Innovation propensity” is also *p* < 0.1.

Among the socio-demographic variables, the education level, employment status, and the presence of an elderly person in the household have shown significance at the 0.1, 0.05, and 0.01 significance levels, respectively. Table 7 displays the β and odds ratio (OR) values calculated for the education level and knowledge level variables. The results indicate that for consumers with a higher education level, their knowledge or lack of knowledge regarding the nutritional content of potatoes does not alter the influence. In both cases, β > 0 and OR > 1 indicate that a higher education level is associated with a significantly increased propensity to purchase potatoes from supermarkets. Similar results are observed for employed respondents, as they exhibit a 112.2% higher likelihood of preferring supermarkets compared to the unemployed. These findings suggest that, in addition to monetary resources and education, there may be other variables influencing the choice of purchasing channel. Examining the OR values, the study found that individuals who perceive potatoes as a carbohydrate and those who believe that potatoes require a long cooking time are 22.4% and 13.8% more likely, respectively, to prefer supermarkets compared to those who do not hold these opinions. However, different results are observed when considering a high knowledge level in the context of a low education level. In this case, both the β and OR values indicate a 48.8% increase in the likelihood of preferring proximity channels. It is interesting to note that consumers who are more inclined to shop at local channels are aware of the nutritional potential of potatoes, primarily due to their actual knowledge of the topic rather than solely relying on a higher education level. These findings support the notion that proximity channels provide an ideal platform for launching marketing campaigns that emphasize the nutritional potential of potatoes.

Consumers who have elderly people in their households show a 67.1% increased likelihood of choosing proximity channels, highlighting the importance of time availability as a significant factor. Additionally, the results indicate that consumers who prefer proximity channels tend to have a lower innovation propensity and a preference for more traditional potato varieties. On the other hand, a unit increase in the innovation propensity index corresponds to a 17% higher probability of being inclined towards supermarkets. This suggests that supermarket consumers are often practical and efficient buyers who value pragmatism and time management and are open to trying new and innovative products. Furthermore, the findings support the idea that consumers who have increased their potato consumption in the last five years have a 56.8% higher likelihood of preferring supermarkets compared to those who have not changed their consumption habits. This could be attributed to the higher availability of pre-cooked, processed, and innovative potato products found in supermarkets.

The pseudo-R2 shows that 18.8% of the variance is explained by the model, which has a correct predictive ability of 67.7%.

## 4. Discussion

The role of diets in fostering the transition to sustainable and healthy food systems has gained recognition from scientific bodies, and increasing potato consumption is seen as a positive dietary change that should be adopted [27,28]. In Italy, potatoes hold an important position within local food systems, with over fifty potato varieties officially recognized by the Italian government as typical products due to their historical and geographical reference and six potato varieties with EU-recognized designations of origin and geographical indications [29]. However, potato consumption in Italy has been experiencing a gradual decline. Italian consumers show confusion regarding the nutritional profile of potatoes, with approximately 34% considering them “diet enemies” [19]. Recent evidence indicates a decrease in potato consumption among Italian consumers. In 2022, sales of fresh potatoes accounted for only 6% of total consumer expenditure on vegetables. In addition, in 2022, Italian consumers purchased 2.8% fewer potatoes in volume compared to 2021 [30]. In light of these trends, it is crucial to implement targeted marketing strategies to revitalize the potato market. To achieve this, a comprehensive understanding of consumers’ attitudes and perspectives on potatoes is essential. Consequently, the identification of consumer potato-purchasing outlets’ behaviors and attitudes can assist farmers, food companies, and retailers in defining target groups for their potato products [31].

This study focuses on the choice of the potato-selling channel, which is becoming increasingly vital for product success. The aim is to establish consumer segmentation based on preferred potato purchasing channels. The research identifies two main consumer groups. Firstly, there is a group of consumers who predominantly purchase potatoes from supermarkets. This group comprises individuals with higher education levels, varying degrees of knowledge about potatoes, employment, increased potato consumption, and a perception of potatoes primarily as a carbohydrate requiring a lengthy cooking time. This group also displays an interest in potato innovations. Secondly, there is a group of consumers who prefer proximity channels for potato purchases. This group includes households with older individuals and consumers with lower levels of education but a higher level of knowledge about the nutritional characteristics of potatoes.

### 4.1. Socio Demographic Characteristics

Past studies focused on the influence of socio-demographic variables in the choice of shopping channels reached controversial results [32], with some studies defining them as weak discriminators compared to attitudinal variables [23,33]. Nevertheless, they are still frequently applied in consumer segmentation due to their role in the consumer’s decision-making process and because they are easily accessible variables [31,34,35]. The current study has identified education level and employment situation as key socio-demographic characteristics. The academic education level and being employed increase consumers’ preference for supermarkets. In contrast with these findings, several authors define that, when fruit and vegetables are considered, both educational levels and income increases are usually associated with a lower propensity to shop at supermarkets, mainly due to the perceived higher quality of these items in proximity channels [34,35]. However, potatoes are a particular type of horticultural product, often perceived by consumers as a commodity and as highly homogeneous across purchasing channels [19]. For this reason, the underlying factor that influences the choice of this specific product may be convenience, defined in terms of time management and valorization [32,36]. Nowadays, time convenience is becoming increasingly central in determining the purchasing channel in which to shop, as evidenced by the rise of online shopping [37]. These elements are also consistent with the results presented by the current study: people who have increased their potato consumption prefer the supermarket, as it allows them to buy potatoes in a more time-efficient manner. Furthermore, Bannor et al.’s (2022) [20] research supports that food expenditure on unprocessed agricultural products decreases for consumers with higher educational levels as they are more likely to follow modern trends, such as the consumption of processed food. In the context of the current study, educated people tend to prefer the supermarket when shopping for potatoes, as the availability of processed potato products is greater with respect to proximity channels. In addition, the results of this study confirm that households with older people are more likely to purchase potatoes through proximity channels. Accordingly, the inclination to buy unprocessed food increases with age [36]. Finally, the increased preference of households with older individuals for proximity channels further confirms the assumption of time convenience. As older people often have more free time available for cooking and shopping, their inclination to choose slower-paced channels, such as proximity channels, tends to increase.

### 4.2. Attitudes and Beliefs

The choice of purchasing channel for consumers is influenced not only by their socio-demographic characteristics but also by their attitudes and beliefs regarding food consumption [38]. One important factor is consumers’ attitudes toward cooking. When consumers perceive potatoes as requiring a long cooking time, they are more likely to prefer purchasing them from supermarkets. Previous research has shown a positive relationship between consumers’ positive inclination towards cooking and their preference for proximity channels [38]. Furthermore, consumer beliefs about potatoes also impact their purchasing behavior. Around 40% of respondents strongly agree that potatoes are not considered a vegetable but rather a carbohydrate, similar to bread. These individuals are more likely to prefer purchasing potatoes from supermarkets. Previous studies on the Italian potato market have indicated that consumers’ choices are often influenced by preconceived stereotypical beliefs, particularly when they have limited time for grocery shopping and possess low knowledge about the product [19]. Consumers tend to view potatoes as a commodity, perceiving little differentiation across brands and varieties. This is consistent with the findings of the present study, which show that consumers oriented towards supermarkets are convenience driven and have a generally low level of knowledge about potatoes.

### 4.3. Knowledge and Interest

The level of knowledge and interest in a product can play a significant role in consumer purchasing behavior [39]. The results of this study indicate that individuals with low knowledge tend to have a higher preference for supermarkets. Similarly, Pavlić et al. (2020) [19] found that uninterested consumers predominantly choose supermarkets as their preferred purchasing channel. In the present study, respondents with high knowledge about the nutritional properties of potatoes are more inclined to choose proximity channels, especially if they do not have an academic background. Insufficient knowledge is a key factor contributing to consumers’ perception of potatoes as undifferentiated products. In a study by Tirelli (2013) [19], it was observed that many Italians struggle to identify the different potato varieties they consume, as well as their origins, flavors, and best uses. This lack of knowledge negatively affects both proximity channels and product innovations. Both are usually connected with higher prices, which are not appreciated by consumers unless they are well informed about the different production methods and characteristics of the crops [35,39]. Additionally, Soós and Biacs (2018) [39] suggest that the higher the existing knowledge level, the greater the need for information. Therefore, based on the results of this study, it is important for proximity channels to emphasize the health and nutritional benefits associated with each potato variety. This can help increase consumer awareness and appreciation for the unique qualities of different potato varieties they may find.

### 4.4. Innovations

Consumer willingness to purchase is often influenced by additional information about a food product. The timing and placement of advertising campaigns are crucial to effectively targeting the right group of consumers [40]. This is particularly important for product innovations. As mentioned earlier, consumers’ beliefs about a product or process strongly influence their choices. If consumers have a negative perception of or lack trust in innovations or innovative products, marketing efforts focused on innovations may decrease their willingness to try new products. Conversely, if consumers have a positive attitude and are open to trying new products, providing information can enhance product credibility [41]. In this study, the inclination towards innovation was measured by assessing respondents’ willingness to purchase different types of innovative potatoes, such as enriched potatoes, biofortified potatoes, and potatoes with increased nutrients. The results indicate that consumers who are more receptive to innovation tend to prefer shopping at supermarkets for their potato purchases. While the overall credibility of the store is also an important factor when consumers consider purchasing innovative products [42], these findings suggest that producers selling their innovative potato products in supermarkets should emphasize the specific characteristics of their products. At the same time, producers should adopt proximity-selling strategies for potato varieties that are known for their distinctive nutritional qualities. Previous studies have highlighted that by incorporating educational content that highlights the benefits of their products, producers can increase consumer acceptance and trust across all channels [41].

### 4.5. Managerial Implications

By providing insights into the characteristics of Italian consumers in two primary potato-selling channels, supermarkets and proximity channels, this study offers valuable information for retailers and producers to develop effective marketing strategies that enhance the visibility and appeal of potatoes. In the case of supermarket marketing, efforts should be made to challenge the stereotype of potatoes as time-consuming to cook and not suitable for diets. As supermarket consumers are receptive to innovations, one approach could be to promote new potato varieties or highlight the unique qualities of traditional varieties that are often overlooked by supermarket consumers. On the other hand, consumers in proximity channels already possess good knowledge of the nutritional properties of potatoes. To address their unfamiliarity with new potato products, it is important to provide them with more detailed information about innovative production methods and the resulting characteristics of the new potato varieties. [43]. This will enable them to perceive these innovative potatoes as natural and equally appealing as the traditional ones.

### 4.6. Limitations and Future Research

While the existing literature often addresses sales channel strategies from the perspective of producers and retailers, the findings of this study highlight the importance of defining consumers’ purchasing channels and tailoring marketing strategies based on their specific attitudes. However, there are limitations and opportunities for further research related to this study. First, the online survey used in this study allowed for reaching a large number of participants efficiently. However, it may have introduced bias as not all consumers have equal access to the Internet, particularly among elderly individuals or low-income families. Second, the convenience sampling approach employed in this study may limit the generalization of the findings, as the sample may not be representative of the overall population. While the large number of participants justifies the obtained results, caution is necessary when generalizing these findings. Third, the self-declared nature of habit-related questions may not accurately reflect real consumer behavior. It is important to consider the potential discrepancy between self-reported habits and actual behavior. To this extent, future studies may include different product or channel characteristics, such as discounts, promotions, or pricing strategies, as well as time convenience, in influencing the choice between purchasing channels. Future research could consider incorporating these variables into the model. Fourth, as in Italy, potato prices limitedly vary between the examined purchasing channels and generally fall within a low price range. This study did not explore the impact of family income levels. Future research, especially if carried out in other countries, could collect additional data on income levels and household composition to enhance the statistical representativeness of the findings. Finally, it is important to note that this study focused exclusively on potatoes, while consumers’ choice of purchasing channels may also be influenced by their need to purchase multiple food products. Future research could examine the relationships between consumer attitudes, knowledge, and purchasing channel choices for not only potatoes but also other food products to provide results in a broader context.

## 5. Conclusions

Potatoes are crucial for ensuring food security, yet their consumption in Italy is declining, and consumers’ knowledge about their nutritional importance is also diminishing. To revitalize the potato market, a new marketing strategy is necessary, taking into account consumers’ needs and attitudes towards potatoes. The findings of this study have identified two primary consumer segments based on their preferred potato-purchasing channel. Consumers who prefer to buy potatoes from supermarkets are typically highly educated and employed individuals who have increased their potato consumption. They perceive potatoes as carbohydrates rather than vegetables and believe that potatoes require a long cooking time. This group is also interested in potato innovations and exhibits a mix of good and poor knowledge about potatoes. Supermarket potato consumers prioritize timesaving and are receptive to innovations, making them more open to trying new and innovative potato products. On the other hand, consumers who prefer proximity channels for purchasing potatoes are typically older individuals with lower levels of education but a higher level of knowledge about potato characteristics and nutritional values. The role of knowledge appears to be a significant discriminating factor for consumers without academic education, emphasizing the potential impact of well-designed information campaigns on their purchasing habits. 

Based on the research findings, future marketing campaigns should be tailored to the specific retail channel. In supermarkets, campaigns should highlight the innovative aspects of potatoes and aim to change consumers’ perception of potatoes as time-consuming. In proximity channels, marketing efforts should focus on promoting the nutritional benefits of potatoes and educating consumers about innovative potato products. By recognizing the importance of consumer expectations and habits in the food chain, it is possible to develop successful marketing campaigns that cater to their needs and attitudes.

## Figures and Tables

**Table 1 foods-12-02877-t001:** Potatoes micronutrient composition.

	Not Contained 1	Low Content 2	Moderate Content 3	Quite Content 4	High Content 5	I Don’t Know
Starch					32 g	
Fat		2 g				
Potassium			1 g			
Vitamin C				16 mg		
Iron		1 mg				
Protein		4 g				
Sodium (salt)		14 mg				
Cholesterol	-					
Carbohydrates					36 g	
Folate			-			
Vitamin B			-			

Note: The table presents the items that were provided to respondents to assess their knowledge level of potatoes’ micronutrients. Respondents were asked to indicate the approximate content level of each element using a Likert scale ranging from 1 to 5, where “1” represented “Not contained” and “5” represented “High content”. The numbers displayed in the table represent the nutritional values of boiled potatoes as provided by the nutritionist team. These values were calculated based on a 100 g serving size [25].

**Table 2 foods-12-02877-t002:** Socio-demographic analysis of the respondents.

Socio-Demographic Characteristics	% of the Total
Gender	
Male	21.80
Female	78.20
Total	100
Age	
18–34	25.7
35–54	46.5
55–64	20.6
Over 65	7.2
Total	100
Level of education	
Junior high school	21.7
High school	41.6
University diploma or advanced	36.7
Total	100
Employment	
Employed	58
Freelancer	16.5
Student	8.7
Unemployed	5.5
Retired	11.3
Total	100
Elderly people at home	
None	81.6
One or more	18.4
Total	100

**Table 3 foods-12-02877-t003:** Knowledge index description and association with the perceived knowledge.

Knowledge Index		
Mean	6.4	
Standard deviation	1.5	
Quartile 1	5.76	
Quartile 2	6.67	
Quartile 3	7.58	
Min	0.9	
Max	9.7	
		Knowledge index mean
Perceived knowledge	Low	5.9
	Medium	6.7
	High	7.3
χ^2^ = 86.424 with *p* < 0.05		

**Table 4 foods-12-02877-t004:** Willingness to buy innovative potatoes, and innovation propensity index levels description.

Willingness to buy innovative potatoes	% of Positive Answers
Willingness to buy enriched potatoes	64.3
Willingness to buy biofortified potatoes	56.8
Willingness to buy potatoes with increased nutrients	46.2
Innovation propensity index, levels description	% of the Total
0: No propensity	23.5
1: Low propensity	18.1
2: Medium propensity	25.3
3: High propensity	33.1
	100

Note: The innovation propensity index ranges from 0 to 3 and is calculated by summing the positive responses to the questions regarding the willingness to purchase various innovative products. A score of 0 indicates that all the questions received negative answers. A score of 1 indicates that at least one of the answers was positive. A score of 2 suggests that there were at least two positive responses, and a score of 3 indicates that respondents expressed a willingness to purchase all the innovative products.

**Table 5 foods-12-02877-t005:** Purchasing channel variable description.

	% of the Total	Cumulative %
Exclusively supermarket	55.2	55.2
Exclusively proximity	33.1	88.3
Mixed channels, both supermarket and proximity	11.7	100
Total	100	

**Table 6 foods-12-02877-t006:** Logistic regression results.

Variable	β	OR	95% Confidence Intervals	
			Lower	Upper
Purchasing channel (proximity channels)				
Consumption trend (Unchanged)		**		
Increased consumption in the last 5 years	0.450	1.568 *	0.924	2.663
Diminished consumption in the last 5 years	−0.392	0.676	0.409	1.116
Knowledge level (low knowledge)	−0.669	0.512 ***	0.312	0.842
Knowledge level x Educational level (low knowledge and low education level)	−1.366	0.255 ***	0.101	0.645
Innovation propensity (per unit increment)	0.157	1.169 *	0.995	1.374
Believes and attitudes:				
Potatoes have a good quality–price relationship (per unit increment)	0.131	1.140	0.946	1.375
I do not consider potatoes as a vegetable, but as a carbohydrate such as bread (per unit increment)	0.202	1.224 **	1.046	1.432
Potatoes require long cooking time (per unit increment)	0.129	1.138 *	0.986	1.313
Potatoes are healthy (per unit increment)	−0.026	0.974	0.787	1.205
Sex (male)	0.205	1.228	0.780	1.933
Age (54 y.o. or lower)	−0.089	0.915	0.566	1.481
Education level (high school or lower)	0.807	2.240 *	0.996	5.036
Employment (unemployed)	0.752	2.121 **	1.293	3.479
Number of elderly people at home (none)	−1.113	0.329 ***	0.189	0.571
Constant	−0.799	0.450		

Note: Significance levels: *p* < 0.01 ***, *p* < 0.05 **, *p* < 0.1 *. Variables age, education level, employment and knowledge level have been dichotomized for an easier interpretation of the results. The opinions have been collected as Likert scales.

**Table 7 foods-12-02877-t007:** β and OR estimation with interaction.

Logistic Regression Model Estimated Coefficients		
	β	OR
Education level	0.807	2.240
Knowledge level	−0.669	0.512
Education × Knowledge	−1.366	0.255
Education Level = University and more		
Knowledge = low	0.807	2.240
Knowledge = high	0.807–(−1.366) = 2.173	8.784
Knowledge Level = high		
Education level = high school or less	−0.669	0.512
Education level = university and more	−0.669–(−1.366) = 0.697	2.007

Note: The first block presents the β and OR values estimated by the model. The following blocks highlight the coefficient values in different strata. In the second block, the coefficients related to the Knowledge Level are shown while keeping the Education Level constant and equal to “University and more”. In the last block, the coefficients for both Education Levels are displayed while keeping the Knowledge Level constant and equal to “High.

## Data Availability

The data presented in this study are available on request from the corresponding author.

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
