# Peer review of "Consumers’ Attitude towards Supermarket and Proximity Stores as Purchasing Outlets of Italian Potato Consumers"

_foods, 2023, doi:10.3390/foods12152877_

Round 1

Reviewer 1 Report

The paper examined consumers' attitudes toward Italian potatoes, which is quite interesting. The authors also established the background information regarding Italian potatoes quite clearly. However, I didn't find the importance of proximity stores and supermarkets in relation to potatoes. 

Moreover, in the materials section, the intro part before 2.1 can be more precise.  

Another important aspect is the writing. There are inconsistent formatting issues and grammatical mistakes from time to time. Further revision is needed.

Author Response

The paper examined consumers' attitudes toward Italian potatoes, which is quite interesting. The authors also established the background information regarding Italian potatoes quite clearly. However, I didn't find the importance of proximity stores and supermarkets in relation to potatoes. 

Answer:  Thanks for the comment. The authors have strengthened the point of the research relevance in the introduction section.

Moreover, in the materials section, the intro part before 2.1 can be more precise.  Further specification of the survey questions numerosity

Answer: The authors have provided further information on data collection, area of survey dissemination, time of compilation before section 2.1, and in the section explaining the survey question, as suggested by the reviewer.

Reviewer 2 Report

Dear Authors,

The manuscript (foods-24861763) submitted for review is quite interesting and I recommend it after revision.

Authors, Please note and address the following comments:

Keywords have chosen well.

Introduction

The Introduction section is good written.

Material and Methods

How many questions were in the survey? All we know from the manuscript is that there were 6 question sections.

Were all those who wanted to take part in the study could to do, or were there any elements that excluded the respondents from the study, e.g. they did not eat or no buy potatoes?

Discussion

In the text of the manuscript states: „Recent evidence have also highlighted a decline in potato consumption by Italian consumers: in 2022 the consumption volume of fresh potatoes diminished by 2.8% compared to 2021 [35].”

It would be better to write what the level of consumption was in kg/year/capita, because without this reference, I don't know if the 2.8% is a big change or a small one. Perhaps the reduced consumption of fresh potatoes is related to the increased consumption of processed potatoes.

Chapter 3.4 Innovation - what kind of innovation in the case of potatoes are we talking about here. Do the authors mean new varieties of potatoes.

References

References are not cited according to journal rules. Publications from MDPI provide information on how to properly cite. Authors may also find this information in the authors' guide.

 Despite my comments, I am pleased to recommend this manuscript. I believe that it concerns an important area of research in an international context.

 Reviewer

Author Response

Introduction

The Introduction section is good written.

Answer: thanks for the positive comment.

Material and Methods

How many questions were in the survey? All we know from the manuscript is that there were 6 question sections.

Answer: Following the reviewers’ valuable comment, the authors have enriched the explanation on the question sections’ items.

Were all those who wanted to take part in the study could to do, or were there any elements that excluded the respondents from the study, e.g. they did not eat or no buy potatoes?

Answer: The authors would like to specify that respondents declared to eat potatoes.

Discussion

In the text of the manuscript states: „Recent evidence have also highlighted a decline in potato consumption by Italian consumers: in 2022 the consumption volume of fresh potatoes diminished by 2.8% compared to 2021 [35].”

It would be better to write what the level of consumption was in kg/year/capita, because without this reference, I don't know if the 2.8% is a big change or a small one. Perhaps the reduced consumption of fresh potatoes is related to the increased consumption of processed potatoes.

Answer: Thanks the pointing out the issue. The authors edited the manuscript accordingly to clarify the information on the level of consumption as reported in the reference source used.

Chapter 3.4 Innovation - what kind of innovation in the case of potatoes are we talking about here. Do the authors mean new varieties of potatoes.

Answer: the valuable comment suggested the authors to clarify on the potato innovation issue. Specific questions have been asked in the questionnaire regarding the different kind of innovations, that is enriched potatoes, biofortified potatoes, and potatoes with increased nutrients.

References

References are not cited according to journal rules. Publications from MDPI provide information on how to properly cite. Authors may also find this information in the authors' guide.

Answer: The authors have modified the references as suggested by the reviewers.

Despite my comments, I am pleased to recommend this manuscript. I believe that it concerns an important area of research in an international context.

Answer: the authors would like to thank for the nice comments and constructive review recommendations.

Reviewer 3 Report

foods-2486163

Article Title: Consumers’ attitude towards supermarket and proximity stores as purchasing outlets of Italian potato consumers 

The manuscript deals with a potentially interesting topic albeit in its current form it fails to provide a well-structured justification of the importance and potential contribution of the empirical analysis that is presented. In that respect I would suggest that this is work in progress and major changes are needed. Some major concerns and suggestions that the authors might wish to consider are:

1)      There is no connection between the way potatoes are cooked and consumers’ perceptions of their nutritional value. Given that they cannot be consumed raw the study should account for how potatoes are usually consumed within the household. Then I would suggest that the indicative info would be to connect actual nutritional value of (e.g. French fries or steam cooked) potatoes with perceived nutritional value of French fries or steamed potatoes.

2)      In addition to point 1, the size and composition of the household is an important parameter that should be accounted for in the analysis as the buying and consumption patterns of e.g. students and families with young children are expected to differ.

3)      Also, the study should account for the person who is responsible for cooking and / or shopping as preparation matters and so does the time available for cooking of e.g. a working member of the household and / or a non-working member of the household.

4)      The choice to buy potatoes from a supermarket instead of a convenient store is much related to all of the above and the income of the household, the percentage of family budget that is usually devoted to last minute or unforeseen purchases made in the neighborhoods’ convenient store, or other shopping decisions that for example relate to personal relationships developed with a near buy provider that delivers special (e.g. local producers’) products etc.

5)      At the study level the choice to make an online survey that is not representative of a given population for which conclusions to draw is also problematic.

6)      Finally, I would suggest that the text is corrected by an English speaking editor.  

The text should be corrected by an English speaking editor. 

Author Response

The manuscript deals with a potentially interesting topic albeit in its current form it fails to provide a well-structured justification of the importance and potential contribution of the empirical analysis that is presented. In that respect I would suggest that this is work in progress and major changes are needed. Some major concerns and suggestions that the authors might wish to consider are:

Answer: thanks for the comments. The authors trust the integrations carried out allowed to improve the paper. Moreover, in line with the above comment, the research potential contribution and justification were further clarified.

  • There is no connection between the way potatoes are cooked and consumers’ perceptions of their nutritional value. Given that they cannot be consumed raw the study should account for how potatoes are usually consumed within the household. Then I would suggest that the indicative info would be to connect actual nutritional value of (e.g. French fries or steam cooked) potatoes with perceived nutritional value of French fries or steamed potatoes.

Answer: thanks, we agree with the reviewer and the study now better specifies the type of potato analysed. In particular, the potato used for reference was the boiled potato. This is common in past sensory studies and by nutritionists in the Italian nutritional guidelines, as well as popularity of this cooking method in the Italian culinary tradition.

  • In addition to point 1, the size and composition of the household is an important parameter that should be accounted for in the analysis as the buying and consumption patterns of e.g. students and families with young children are expected to differ.

Answer: Thanks for pointing this out. The research explored the household size and presence of children in the household, and the results were not significant or the available data provided by respondents was not adequate. The researchers included the presence of the elderly as it resulted a statistically significant variable and it was thus included in the model.

  • Also, the study should account for the person who is responsible for cooking and / or shopping as preparation matters and so does the time available for cooking of e.g. a working member of the household and / or a non-working member of the household.

Answer: thanks for the comment. The authors agree that the food cooking and shopping can impact on how households approach food behavior. The study highlights this in relation to households with elderly consumers, that are a non-working household member. To highlight this aspect the paper was adequaly integrated. Furthermore, the above aspect highlighted by the reviewer was suggested as a relevant variable in the limitations and further research section .

4)      The choice to buy potatoes from a supermarket instead of a convenient store is much related to all of the above and the income of the household, the percentage of family budget that is usually devoted to last minute or unforeseen purchases made in the neighborhoods’ convenient store, or other shopping decisions that for example relate to personal relationships developed with a near buy provider that delivers special (e.g. local producers’) products etc.

Answer: The comment is insightful; however, it should be noted that potato prices in Italy are generally low and do not vary significantly across different purchasing channels. Furthermore, income is typically considered sensitive data and despite collected it was not made available by an adequate number of consumers. The authours would like to specificy that given the scope of the study, our investigation focused on categorizing respondents as either working or non-working individuals. As we believe it may be appropriate in other countries, the authors highlighted the point in the limitation section.

In relation to unforeseen purchases, these were not included in the analysis as they are typically driven by immediate and temporary needs, rather than habits and attitudes. Moreover, potatoes are often part of the general shopping list for families, and their purchase is usually planned in advance and possibly made during regular shopping trips. However, it is acknowledged that the need to purchase multiple food products may influence the choice of purchasing channel, thus this point is included in the limitation of the study.

  • At the study level the choice to make an online survey that is not representative of a given population for which conclusions to draw is also problematic.

Answer: the valuable comment suggested to further speficy this aspect in the study. The authors would like to clarify that the sampling approach applied is convenience sampling. The generalization limitation have been highlighted in the limitation section.

5)      Finally, I would suggest that the text is corrected by an English speaking editor.  

Answer: as suggested by the reviewer, the English language was revised

Round 2

Reviewer 3 Report

The text has been improved as a result of the clarifications that were added.

It is my view that the word consumers at the end of the title phrase is redundant. The authors might wish to check that with an english speaking editor.